# GRAPH-BASED CONTINUAL LEARNING

**Binh Tang**
Department of Statistics and Data Science
Cornell University
Ithaca, NY 14850
`bvt5@cornell.edu`

**David S. Matteson**
Department of Statistics and Data Science
Cornell University
Ithaca, NY 14850
`matteson@cornell.edu`

## ABSTRACT

Despite significant advances, continual learning models still suffer from catastrophic forgetting when exposed to incrementally available data from non-stationary distributions. Rehearsal approaches alleviate the problem by maintaining and replaying a small episodic memory of previous samples, often implemented as an array of independent memory slots. In this work, we propose to augment such an array with a learnable random graph that captures pairwise similarities between its samples, and use it not only to learn new tasks but also to guard against forgetting. Empirical results on several benchmark datasets show that our model consistently outperforms recently proposed baselines for task-free continual learning.

## 1 INTRODUCTION

Recent breakthroughs of deep neural networks often hinge on the ability to repeatedly iterate over stationary batches of training data. When exposed to incrementally available data from non-stationary distributions, such networks often fail to learn new information without forgetting much of its previously acquired knowledge, a phenomenon often known as *catastrophic forgetting* (Ratcliff, 1990; McCloskey & Cohen, 1989; French, 1999). Despite significant advances, the limitation has remained a long-standing challenge for computational systems that aim to continually learn from dynamic data distributions (Parisi et al., 2019).

Among various proposed solutions, rehearsal approaches that store samples from previous tasks in an episodic memory and regularly replay them are one of the earliest and most successful strategies against catastrophic forgetting (Lin, 1992; Rolnick et al., 2019). An episodic memory is typically implemented as an array of independent slots; each slot holds one example coupled with its label. During training, these samples are interleaved with those from the new task, allowing for simultaneous multi-task learning as if the resulting data were independently and identically distributed.

While such approaches are effective in simple settings, they require sizable memory and are often impaired by memory constraints, performing rather poorly on complex datasets. A possible explanation is that slot-based memories fail to utilize relational structure between samples; semantically similar items are treated independently both during training and at test time. In marked contrast, relational memory is a prominent feature of biological systems that has been strongly linked to successful memory retrieval and generalization (Prince et al., 2005). Humans, for example, encode event features into cortical representations and bind them together in the medial temporal lobe, resulting in a durable, yet flexible form of memory (Shimamura, 2011).

In this paper, we introduce a novel Graph-based Continual Learning model (GCL) that resembles some characteristics of relational memory. More specifically, we explicitly model pairwise similarities between samples, including both those in the episodic memory and those found in the current task. These similarities allow for representation transfer between samples and provide a resilient mean to guard against catastrophic forgetting. Our contributions are twofold:

(1) We propose the use of random graphs to represent relational structures between samples. While similar notions of dependencies have been proposed in the literature (Louizos et al., 2019; Yao et al., 2020), the application of random graphs in task-free continual learning is novel, at least to the best of our knowledge.

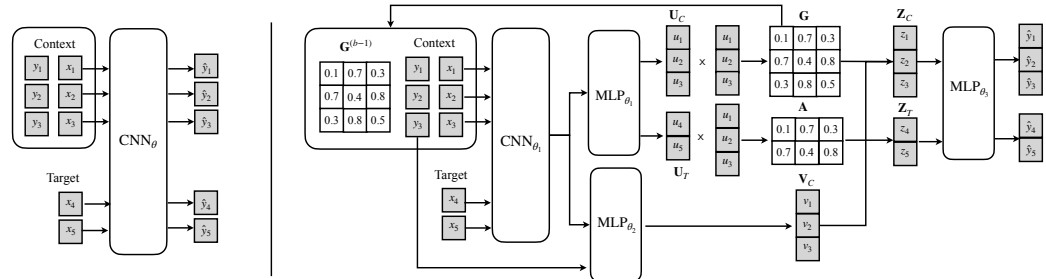

Figure 1: Illustration of Experiment Replay (ER) (Chaudhry et al., 2019) on the *left* and our model (GCL) on the *right*. While ER independently processes *context images* from the episodic memory and *target images* from the current task, GCL models pairwise similarities between the images via the random graphs **G** and **A**.

(2) We introduce a new regularization objective that leverages such random graphs to alleviate catastrophic forgetting. In contrast to previous work (Rebuffi et al., 2017; Li & Hoiem, 2017) based on knowledge distillation (Hinton et al., 2015), the objective penalizes the model for forgetting learned edges between samples rather than their output predictions.

Our approach performs competitively on four commonly used datasets, improving accuracy by up to 19.7% and reducing forgetting by almost 37% in the best case when bench-marked against competitive baselines in task-free continual learning.

## 2 PROBLEM FORMULATION

In this work, we follow the learning protocol for image classification from Lopez-Paz & Ranzato (2017). More specifically, we consider a training set $\mathcal{D} = \{\mathcal{D}_1, \cdots, \mathcal{D}_T\}$ consisting of $T$ tasks where the dataset for the $t$-th task $\mathcal{D}_t = \{(\mathbf{x}_i^t, \mathbf{y}_i^t)\}_{i=1}^{n_t}$ contains $n_t$ input-target pairs $(\mathbf{x}_i^t, \mathbf{y}_i^t) \in \mathcal{X} \times \mathcal{Y}$. While the tasks arrive sequentially and exclusively, we assume the input-target pairs $(\mathbf{x}_i^t, \mathbf{y}_i^t)$ in each task are independent and identically distributed (i.i.d.). The goal is to learn a supervised model $f_\theta : \mathcal{X} \to \mathcal{Y}$, parametrized by $\theta$, that outputs a class label $\mathbf{y} \in \mathcal{Y}$ given an unseen image $\mathbf{x} \in \mathcal{X}$.

Following prior work (Lopez-Paz & Ranzato, 2017; Riemer et al., 2018; Chaudhry et al., 2019), we consider online streams of tasks in which samples from different tasks arrive at different times. As an additional constraint, we insist that the model can only revisit a small amount of data chosen to be stored in a fixed-size episodic memory $\mathcal{M}$.

For clarity, we refer to the data in such an episodic memory as *context images* and *context labels* and denote by $\mathbf{X}_\mathcal{C} = \{\mathbf{x}_i\}_{i \in \mathcal{C}}$ and $\mathbf{Y}_\mathcal{C} = \{\mathbf{y}_i\}_{i \in \mathcal{C}}$, respectively. These images and labels are to be distinguished from those in the current task, which we refer to as *target images* and *target labels* and denote by $\mathbf{X}_\mathcal{T} = \{\mathbf{x}_j\}_{j \in \mathcal{T}}$ and $\mathbf{Y}_\mathcal{T} = \{\mathbf{y}_j\}_{j \in \mathcal{T}}$, respectively. While the model is allowed to update the context samples during training, the episodic memory is necessarily frozen at test time.

## 3 GRAPH-BASED CONTINUAL LEARNING

In this section, we propose a Graph-based Continual Learning (GCL) algorithm. While most rehearsal approaches ignore the correlations between images and independently pass them through a network to compute predictions (Rebuffi et al., 2017; Chaudhry et al., 2019; Aljundi et al., 2019c), we model pairwise similarities between the images with learnable edges in random graphs (see Figure 1). Intuitively, although it might be easy for the model to forget any particular sample, the multiple connections it forms with similar neighbors are harder to be forgotten altogether. If trained well, the random graphs can therefore equip the model with a plastic and durable means to fight against catastrophic forgetting.

**Graph Construction.** Given a minibatch of target images $\mathbf{X}_\mathcal{T}$ from the current task, our model makes predictions based on the context images $\mathbf{X}_\mathcal{C}$ and context labels $\mathbf{Y}_\mathcal{C}$ that span several previously seen tasks, up to and including the current one. In particular, we explicitly build two random graphs

of pairwise dependencies: an undirected graph $\mathbf{G}$ between the context images $\mathbf{X}_{\mathcal{C}}$ and a directed, bipartite graph $\mathbf{A}$ from the context images $\mathbf{X}_{\mathcal{C}}$ to the target images $\mathbf{X}_{\mathcal{T}}$.

Since an undirected graph can be thought of as a directed graph between its vertices and a copy of itself, we treat the *context graph* $\mathbf{G}$ as such and build it analogously to the *context-target graph* $\mathbf{A}$. Specifically, the high-dimensional context images $\mathbf{X}_{\mathcal{C}}$ and target images $\mathbf{X}_{\mathcal{T}}$ are first mapped to the image embeddings $\mathbf{U}_{\mathcal{C}}$ and $\mathbf{U}_{\mathcal{T}}$, respectively, using an image encoder $f_{\theta_1} : \mathcal{X} \rightarrow \mathbb{R}^{d_1}$. Following Louizos et al. (2019), we then represent the edges in each graph by independent Bernoulli random variables whose means are specified by a kernel function in the embedding space. More precisely, the distribution of the resulting Erdős-Rényi random graphs (Erdős & Rényi, 1959) can be defined as

$$p(\mathbf{G} \,|\, \mathbf{U}_{\mathcal{C}}) = \prod_{i \in \mathcal{C}} \prod_{k \in \mathcal{C}} \text{Ber}(\mathbf{G}_{ik} \,|\, \kappa_{\tau}(\mathbf{u}_i, \mathbf{u}_k)), \tag{1}$$

$$p(\mathbf{A} \,|\, \mathbf{U}_{\mathcal{T}}, \mathbf{U}_{\mathcal{C}}) = \prod_{j \in T} \prod_{k \in \mathcal{C}} \text{Ber}(\mathbf{A}_{jk} \,|\, \kappa_{\tau}(\mathbf{u}_j, \mathbf{u}_k)), \tag{2}$$

for all $i, k \in \mathcal{C}$ and $j \in \mathcal{T}$ where $\kappa_{\tau} : \mathbb{R}^{d_1} \times \mathbb{R}^{d_1} \rightarrow [0, \infty)$ is a kernel function that encodes similarities between image embeddings such as the RBF kernel $\kappa_{\tau}(\mathbf{u}_i, \mathbf{u}_j) = \exp\left(-\frac{\tau}{2}\|\mathbf{u}_i - \mathbf{u}_j\|_2^2\right)$. Here, with a slight abuse of notation, we also use $\mathbf{G}$ and $\mathbf{A}$ to denote the corresponding adjacency matrices; $\mathbf{A}_{jk} \in \{0, 1\}$, for example, represents the presence or absence of a directed edge between the $j$-th target image and the $k$-th context image.

**Predictive Distribution.**   Given a context graph $\mathbf{G}$ and a context-target graph $\mathbf{A}$ that encode pairwise similarities to the context images, our next step is to propagate information from the context images $\mathbf{X}_{\mathcal{C}}$ and context labels $\mathbf{Y}_{\mathcal{C}}$ to make predictions. To that end, we embed $\mathbf{X}_{\mathcal{C}}$ by another image encoder $f_{\theta_2}$ with weights partially tied to the previous one $f_{\theta_1}$, and encode $\mathbf{Y}_{\mathcal{C}}$ by a linear label encoder before concatenating the resulting embeddings into latent representations $\mathbf{V}_{\mathcal{C}} \in \mathbb{R}^{|\mathcal{C}| \times d_2}$. In combination with the distributions of $\mathbf{G}$ and $\mathbf{A}$, we compute context-aware representations for the context images and target images, denoted by $\{\mathbf{z}_i\}_{i \in \mathcal{C}}$ and $\{\mathbf{z}_j\}_{j \in \mathcal{T}}$, respectively:

$$p(\mathbf{z}_i \,|\, \mathbf{U}_{\mathcal{C}}, \mathbf{V}_{\mathcal{C}}) = \int_{\mathbf{G}} \mathbb{I}_{\{\tilde{\mathbf{G}}_i \mathbf{V}_{\mathcal{C}}\}}(\mathbf{z}_i) \, dP(\mathbf{G} \,|\, \mathbf{U}_{\mathcal{C}}) \tag{3}$$

$$p(\mathbf{z}_j \,|\, \mathbf{U}_{\mathcal{T}}, \mathbf{U}_{\mathcal{C}}, \mathbf{V}_{\mathcal{C}}) = \int_{\mathbf{A}} \mathbb{I}_{\{\tilde{\mathbf{A}}_j \mathbf{V}_{\mathcal{C}}\}}(\mathbf{z}_j) \, dP(\mathbf{A} \,|\, \mathbf{U}_{\mathcal{T}}, \mathbf{U}_{\mathcal{C}}). \tag{4}$$

where $\tilde{\mathbf{G}}_i$ and $\tilde{\mathbf{A}}_j$ indicate the $i$-th and $j$-th row of $\mathbf{G}$ and $\mathbf{A}$, each normalized to sum to 1, and $\mathbb{I}_{\mathcal{S}}(\cdot)$ denotes the indicator function on a set $\mathcal{S}$. Intuitively, the representations $\mathbf{V}_{\mathcal{C}}$ are linearly weighted by each graph sample, and the normalization step ensures proper scaling in case the numbers of edges formed with the context images vary. Once we summarize each image by the context samples, a final network $f_{\theta_3} : \mathbb{R}^{d_2} \rightarrow \mathcal{Y}$ takes as input the context-aware representations and produces predictive distributions:

$$p(\mathbf{y}_i \,|\, \mathbf{X}_{\mathcal{C}}) = \int_{\mathbf{z}_i} p\left(\mathbf{y}_i \,|\, f_{\theta_3}(\mathbf{z}_i)\right) \, dP(\mathbf{z}_i \,|\, \mathbf{U}_{\mathcal{C}}, \mathbf{V}_{\mathcal{C}}), \tag{5}$$

$$p(\mathbf{y}_j \,|\, \mathbf{x}_j, \mathbf{X}_{\mathcal{C}}) = \int_{\mathbf{z}_j} p\left(\mathbf{y}_j \,|\, f_{\theta_3}(\mathbf{z}_j)\right) \, dP(\mathbf{z}_j \,|\, \mathbf{U}_{\mathcal{T}}, \mathbf{U}_{\mathcal{C}}, \mathbf{V}_{\mathcal{C}}). \tag{6}$$

Since the numbers of random binary graphs $\mathbf{G}$ and $\mathbf{A}$ are exponential, we approximate the integrals in (1) - (6) by Monte Carlo samples. More specifically, we use one sample of $\mathbf{G}$ and $\mathbf{A}$ during training and 30 samples of $\mathbf{A}$ during testing. Also, these graph samples are inherently non-differentiable, so we use the Gumbel-Softmax relaxations of the Bernoulli random variables during training (Maddison et al., 2016; Jang et al., 2016). The degree of approximation is controlled by temperature hyperparameters, which exert significant influence over the density of the graph samples. We find that a small temperature for $\mathbf{G}$ and a larger temperature for $\mathbf{A}$ work well.

There are several reasons for making the graphs $\mathbf{G}$ and $\mathbf{A}$ random. First, the stochasticity induced by the Bernoulli random variables allows us to output multiple predictions and average these predictions, and such ensemble techniques have been quite successful in continual learning settings (Coop et al., 2013; Fernando et al., 2017). Perhaps more importantly, we find that the deterministic version with the Bernoulli random variables replaced by their parameters results in very sparse graphs where

samples from the same classes are often deemed dissimilar. In a similar fashion to dropout (Srivastava et al., 2014), the random edges encourage the model to be less reliant on a few particular edges and therefore promote knowledge transfer between samples. By a similar reasoning, we remove self-edges in the context graph and also observe more connections between samples.

**Graph Regularization.** As training switches to new tasks, the distributional shifts to the target images necessarily result in changes to both the context graph $\mathbf{G}$ and the context-target graph $\mathbf{A}$. In addition, the context images are regularly updated to be representative of the data distribution up to that point, so any well-learned connections between the context images are also susceptible to catastrophic forgetting. As a remedy, we save the parameters of the Bernoulli edges to the episodic memory in conjunction with the context images and context labels, and introduce a regularization term that discourages the model from forgetting previously learned edges:

$$\mathcal{L}_{\mathbf{G}}^{(b)}(\theta_1) \triangleq \frac{1}{|\mathcal{I}^{(b)}|} \ell\Big(p\Big(\mathbf{G}_{\mathcal{I}^{(b)}}^{(b-1)}\Big), \, p\Big(\mathbf{G}_{\mathcal{I}^{(b)}}^{(b)}\Big)\Big). \tag{7}$$

Here, $\ell(\cdot, \cdot)$ denotes the cross-entropy between two probability distributions, $\mathcal{I}^{(b)}$ the index set of edges to be regularized in the $b$th minibatch, and $\mathbf{G}^{(b-1)}$ the adjacency matrix learned from the beginning up to the previous minibatch. The selection strategies $\mathcal{I}^{(b)}$ are discussed in the next subsection. Besides the regularization term, our training objective includes two other cross-entropy losses, one for the context images and another for the target images:

$$\mathcal{L}(\theta_1, \theta_2, \theta_3) = \frac{\lambda_{\mathcal{C}}}{|\mathcal{C}|} \sum_{i \in \mathcal{C}} \ell\Big(\mathbf{y}_i, \hat{\mathbf{y}}_i^{(s)}\Big) + \frac{\lambda_{\mathcal{T}}}{|\mathcal{T}|} \sum_{j \in \mathcal{T}} \ell\Big(\mathbf{y}_j, \hat{\mathbf{y}}_j^{(s)}\Big) + \lambda_{\mathbf{G}} \mathcal{L}_{\mathbf{G}}^{(b)}(\theta_1), \tag{8}$$

where $\hat{\mathbf{y}}_i^{(s)} = f_{\theta_3}(\mathbf{z}_i^{(s)})$, $\hat{\mathbf{y}}_j^{(s)} = f_{\theta_3}(\mathbf{z}_j^{(s)})$ and $\mathbf{z}_i^{(s)} \sim p(\mathbf{z}_i \,|\, \mathbf{U}_{\mathcal{C}}, \mathbf{V}_{\mathcal{C}})$, $\mathbf{z}_j^{(s)} \sim p(\mathbf{z}_j \,|\, \mathbf{U}_{\mathcal{T}}, \mathbf{U}_{\mathcal{C}}, \mathbf{V}_{\mathcal{C}})$ are context-aware samples from Equations 3 and 4, and $\lambda_{\mathcal{C}}, \lambda_{\mathcal{T}}, \lambda_{\mathbf{G}}$ are hyperparameters.

While the graph regularization term appears similar to knowledge distillation (Hinton et al., 2015), we emphasize that the former aims to preserve the covariance structures between the outputs of the image encoder $f_{\theta_1}$ rather than the outputs themselves. We believe that in light of new data, the image encoder should be able to update its potentially superficial representations of previously seen samples as long as it keeps the correlations between them unchanged. Indeed, some of the early regularization approaches based on knowledge distillation (Li & Hoiem, 2017; Rebuffi et al., 2017) are sometimes too restrictive and reportedly underperform in certain scenarios (Kemker & Kanan, 2017).

**Task-Free Knowledge Consolidation.** When task identities are not available, we use reservoir sampling (Vitter, 1985) to update the context images and context labels as in Riemer et al. (2018). The sampling strategy takes as input a stream of data and randomly replaces a context sample in the episodic memory with a target sample, with probability proportional to the number of samples observed so far. Despite its simplicity, reservoir sampling has been shown to yield strong performance in recent work (Chaudhry et al., 2019; Riemer et al., 2018; Rolnick et al., 2019).

While most prior work uses task boundaries to perform knowledge consolidation at the end of each task (Kirkpatrick et al., 2017; Rebuffi et al., 2017), we update the context graph in memory after every minibatch of training data. In addition, such updates are performed at the sample level to maximize flexibility; we keep track of the cross entropy loss on each context sample and only update its edges in the graph when the model reaches a new low (denoted by $\mathcal{I}^{(b)}$ previously). Intuitively, the loss measures how well the model has learned the context image through the connections it forms with others, so meaningful relations are most likely obtained at the bottom of the loss surface. Though samples from the same task often provide more support for each other, the task-agnostic mechanism for updating the context graph also allows for knowledge transfer across tasks when necessary.

**Memory and Time Complexity.** The inclusion of pairwise similarities and graph regularization result in a time and memory complexity of $\mathcal{O}(|\mathcal{M}|^2 + |\mathcal{M}|N)$ and $\mathcal{O}(|\mathcal{M}|^2)$, respectively, where $|\mathcal{M}|$ denotes the size of the episodic memory and $N$ the batch size for target images. The quadratic costs in $|\mathcal{M}|$, however, are not concerning in practice, as we deliberately use a small, fixed-size episodic memory. The cost of storing $\mathbf{G}$ is often dwarfed by the memory required for storing high-dimensional images, as each edge only needs one floating point number (see Appendix E for more details on memory usage).

## 4 RELATED WORK

**Continual Learning Approaches.**    The existing work on continual learning mostly falls into three categories: *regularization*, *expansion*, and *rehearsal*. *Regularization* approaches alleviate catastrophic forgetting by penalizing changes in model weights that are important for past tasks. Different measures of weight importance are considered, including Fisher information (Kirkpatrick et al., 2017; Chaudhry et al., 2018a), synaptic relevance (Zenke et al., 2017), and uncertainty estimates (Ebrahimi et al., 2019). The constraints on weight updates can also be studied from Bayesian perspectives, where the posterior distribution of the weights is approximated and used as the prior for the next task (Nguyen et al., 2017; Ritter et al., 2018; Titsias et al., 2019). These regularization methods are efficient in memory and computational usage but suffer from brittleness due to representation drift (Titsias et al., 2019).

*Expansion* approaches dynamically allocate additional task-specific neural resources as more tasks arrive. Rusu et al. (2016), for example, blocks changes to parameters learned for previous tasks and expands sub-networks while Yoon et al. (2017) performs neuron splitting or duplication upon arrival of new tasks. Recently, non-parametric Bayesian approaches use Dirichlet process mixture models to expand a set of neural networks in a principled way (Jerfel et al., 2019; Lee et al., 2020). By design, these dynamic architectures prevent forgetting but quickly result in considerable model complexity.

Instead of growing model capacity, *rehearsal* approaches maintain a small episodic memory of previous data or, alternatively, train a generative model to produce pseudo-data for past tasks, which are then replayed and interleaved with samples from the new task. Such generative models (Shin et al., 2017; Kemker & Kanan, 2017; Achille et al., 2018; Caccia et al., 2019; Ostapenko et al., 2019) reduce working memory effectively, but they are also susceptible to catastrophic forgetting and invoke the complexity of the generative task (Parisi et al., 2019). In contrast, episodic memory approaches are simpler and remarkably effective against forgetting (Rolnick et al., 2019; Wu et al., 2019). Lopez-Paz & Ranzato (2017) and Chaudhry et al. (2018b), for example, use an episodic storage of past data to impose inequality constraints on gradient updates while Rebuffi et al. (2017) constructs exemplars for knowledge distillation and nearest neighbor search. Recently, it has been shown that simple replay techniques and optimization-based meta-learning on the episodic memory outperform many previous approaches in online settings (Hayes et al., 2019; Chaudhry et al., 2019; 2020; Riemer et al., 2018). Our model is also based on experience replay, but it differs from the other approaches in the way the episodic memory is handled.

**Task-Free Continual Learning.**    In real-world scenarios, task changes are often unknown and definitive boundaries between tasks do not always exist. However, most methods mentioned above rely on explicit task identities or task boundaries to consolidate knowledge or select sub-modules for task adaptation. Despite its significance, there are only a few works that address task-free continual learning. While Aljundi et al. (2019b) heuristically detects peaks in the loss surface to consolidate knowledge, Aljundi et al. (2019c;a) remove the need for task boundaries by a sample selection strategy for the episodic memory. Recently, the aforementioned non-parametric approaches train density estimators to detect task boundaries and perform model expansion (Lee et al., 2020; Rao et al., 2019). In contrast, our approach uses reservoir sampling (Vitter, 1985) to update the episodic memory, similar to Riemer et al. (2018); Chaudhry et al. (2019).

**Learning with Random Graphs.**    Although widely studied in graph theory (West et al., 2001), random graphs appear sparingly in the machine learning literature, perhaps more noticeably in neural architecture search (Xie et al., 2019). Our work is mostly related to previous work on functional neural process (Louizos et al., 2019), where the authors build random graphs of dependencies to represent relational structures between context points in a stochastic process. Our approach is different in that (1) the random graphs are undirected and grow incrementally, (2) no variational inference is required, and (3) it addresses catastrophic forgetting and performs well under continual learning settings.

**Attention Mechanism.**    While we motivate our approach from a graphical perspective, one can also consider it as some form of attention mechanism. In particular, the context graph $\mathbf{G}$ represents self-attention (Vaswani et al., 2017) across context images, and the context-target graph $\mathbf{A}$ represents cross-attention (Bahdanau et al., 2014) between context images and target images. Though advanced mechanisms such as multi-head attention have been applied successfully in many stationary settings (Vaswani et al., 2017; Xu et al., 2015; Zhang et al., 2018; Kim et al., 2019; Sprechmann et al.,

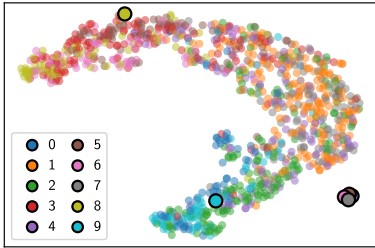 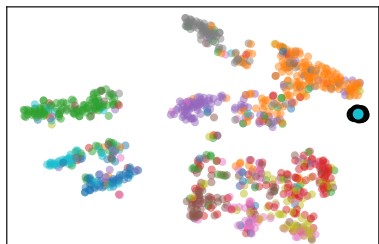

Figure 2: *t*-SNE visualization of image embeddings (small circles) from the penultimate layers and class embeddings (large circles) from the weights of the last layers on SPLIT SVHN. The *left* figure shows that **Finetune**, a model naively trained on the data stream, fails to recognize the class-based clustering structure and bias the image embeddings toward the last task (class 8 & 9). In contrast, the *right* figure shows that **GCL** (our model) maintains the relational structure and is more robust to the distributional shifts incurred by task changes.

2018), we note that naive applications of such techniques in online continual learning suffer from catastrophic forgetting due to representation drift when training switches to new tasks. In contrast, our model employs random attention, which arguably makes it more robust to such distributional shifts (see Figure 2).

## 5 EXPERIMENTS

In this section, we evaluate the proposed GCL model on commonly used continual learning benchmarks. Additional results and details about the datasets, experiment setup, model architectures, and result analyses are available in the appendices.

**Experiment Setup.** We perform experiments on 6 image classification datasets: PERMUTED MNIST, ROTATED MNIST (LeCun et al., 1998), SPLIT SVHN (Netzer et al., 2011), SPLIT CIFAR10 (Krizhevsky et al., 2009), SPLIT CIFAR100 (Krizhevsky et al., 2009), and SPLIT MINIIMAGENET (Vinyals et al., 2016). For each dataset, we follow Lopez-Paz & Ranzato (2017); Chaudhry et al. (2018b) and adopt the setting where the model only has access to an online stream of data with a batch size of 10 (see Appendix A for more details).

We consider both single-head and multiple-head settings. More specifically, we use single-head and one-epoch settings for our model and all baselines on PERMUTED MNIST, ROTATED MNIST, SPLIT SVHN, and SPLIT CIFAR10. While most of previous work (Rebuffi et al., 2017; Lopez-Paz & Ranzato, 2017; Chaudhry et al., 2019) assume task identities on SPLIT CIFAR10, we require all models to perform 10-way classification on each task with the same output head. This variant is more practical and challenging due to the need for incremental knowledge consolidation across tasks.

In addition, we also report results for multiple-head and 10-epochs settings on SPLIT CIFAR100 and SPLIT MINIIMAGENET, following Lopez-Paz & Ranzato (2017). These datasets have more classes and fewer samples per class, rendering them too challenging for single-head settings.

**Model Architecture.** Our image encoders $f_{\theta_1}$ and $f_{\theta_2}$ partially share weights and are parametrized by an MLP on the MNIST variants and a simple 6-layer convolutional network on other datasets, each followed by a RELU activation and a separate linear mapping. As alluded earlier, we use an RBF kernel to compute similarities between image embeddings and find it sufficiently easy for initialization. The output mappings $f_{\theta_3}$ are MLPs in all cases (see Appendix B for more details).

**Baselines.** We benchmark our model against multiple models, including (1) *Finetune*, a popular baseline, naively trained on the data stream; (2) *EWC* (Kirkpatrick et al., 2017), an early regularization approach; (3) GEM (Lopez-Paz & Ranzato, 2017), a rehearsal approach based on an episodic memory of parameter gradients; (4) ER (Chaudhry et al., 2019), a simple yet competitive experience method based on reservoir sampling; (5) MER (Riemer et al., 2018), a rehearsal approach inspired by optimization-based meta-learning, and (6) ICARL (Rebuffi et al., 2017) another well-known rehearsal strategy. Most of these baselines share the same model architectures: an MLP with two hidden layers on the MNIST variants, and a ResNet-18 (He et al., 2016) on SPLIT SVHN and SPLIT CIFAR10, following (Lopez-Paz & Ranzato, 2017) (see Appendix C for more details).

Table 1: Classification results (%) on PERMUTED MNIST, ROTATED MNIST and SPLIT SVHN. The means and standard deviations are computed over five runs using different random seeds, When used, episodic memories contain 5 samples per class on average. The symbol ↑ (↓) indicates that a higher (lower) number is better.

| DATASET | PERMUTED MNIST | | ROTATED MNIST | | SPLIT SVHN | |
|---|---|---|---|---|---|---|
| Method | ACC (↑) | FGT (↓) | ACC (↑) | FGT (↓) | ACC (↑) | FGT(↓) |
| Finetune | $60.19 \pm 2.31$ | $23.62 \pm 1.98$ | $43.80 \pm 1.64$ | $46.52 \pm 1.71$ | $18.85 \pm 0.10$ | $94.78 \pm 1.24$ |
| EWC | $64.94 \pm 1.22$ | $18.33 \pm 1.07$ | $44.99 \pm 1.73$ | $44.98 \pm 1.95$ | $18.76 \pm 0.27$ | $94.99 \pm 1.23$ |
| GEM | $79.17 \pm 0.70$ | $3.68 \pm 0.68$ | $82.60 \pm 0.48$ | $5.47 \pm 0.45$ | $33.40 \pm 3.27$ | $68.91 \pm 4.06$ |
| ER | $79.90 \pm 0.46$ | $3.78 \pm 0.45$ | $80.82 \pm 0.68$ | $6.78 \pm 0.69$ | $45.41 \pm 3.03$ | $62.37 \pm 4.33$ |
| MER | $79.68 \pm 0.42$ | $3.47 \pm 0.41$ | $83.56 \pm 0.23$ | $8.14 \pm 0.46$ | - | - |
| **GCL** | $\mathbf{82.36} \pm 0.36$ | $\mathbf{2.92} \pm 0.23$ | $\mathbf{86.37} \pm 0.32$ | $\mathbf{3.22} \pm 0.50$ | $\mathbf{60.68} \pm 1.67$ | $\mathbf{21.86} \pm 2.35$ |

Table 2: Classification results (%) on SPLIT CIFAR10 and SPLIT CIFAR100 and SPLIT MINIIMAGENET. The means and standard deviations are computed over five runs using different random seeds, When used, episodic memories contain 5 samples per class on average. The symbol ↑ (↓) indicates that a higher (lower) number is better.

| DATASET | SPLIT CIFAR10 | | SPLIT CIFAR100 | | SPLIT MINIIMAGENET | |
|---|---|---|---|---|---|---|
| Method | ACC (↑) | FGT (↓) | ACC (↑) | FGT (↓) | ACC (↑) | FGT (↓) |
| Finetune | $18.46 \pm 0.12$ | $86.48 \pm 1.02$ | $55.39 \pm 1.94$ | $25.94 \pm 1.89$ | $37.84 \pm 0.87$ | $31.41 \pm 1.57$ |
| EWC | $18.49 \pm 0.13$ | $86.95 \pm 1.15$ | $55.60 \pm 1.11$ | $23.53 \pm 1.19$ | $36.61 \pm 2.06$ | $28.17 \pm 4.49$ |
| ICARL | - | - | $58.08 \pm 1.44$ | $24.22 \pm 1.35$ | - | - |
| GEM | $22.88 \pm 3.41$ | $76.90 \pm 5.53$ | $65.66 \pm 0.70$ | $15.52 \pm 0.41$ | $54.06 \pm 0.22$ | $13.17 \pm 0.74$ |
| ER | $29.94 \pm 3.08$ | $72.64 \pm 4.88$ | $69.40 \pm 1.21$ | $11.25 \pm 1.24$ | $58.74 \pm 0.74$ | $9.02 \pm 2.49$ |
| **GCL** | $\mathbf{49.62} \pm 1.85$ | $\mathbf{35.69} \pm 3.33$ | $\mathbf{74.51} \pm 0.99$ | $\mathbf{6.54} \pm 1.26$ | $\mathbf{61.54} \pm 0.57$ | $\mathbf{6.10} \pm 2.73$ |

**Metrics.** Following Lopez-Paz & Ranzato (2017); Chaudhry et al. (2018a; 2019), we evaluate the models using two classification metrics, namely, *average accuracy* and *average forgetting*:

$$\text{ACC} \triangleq \frac{1}{T}\sum_{i=1}^{T} R_{T,i}, \quad \text{FGT} \triangleq \frac{1}{T-1}\sum_{j=1}^{T-1}(R_{T,i} - R_{i,i}), \tag{9}$$

where $R_{i,j}$ denotes the test accuracy on task $j$ after the model has finished task $i$. Intuitively, the former measures the average test accuracy across all tasks while the latter measures the average decrease between each task's peak accuracy and its accuracy at the end of continual learning.

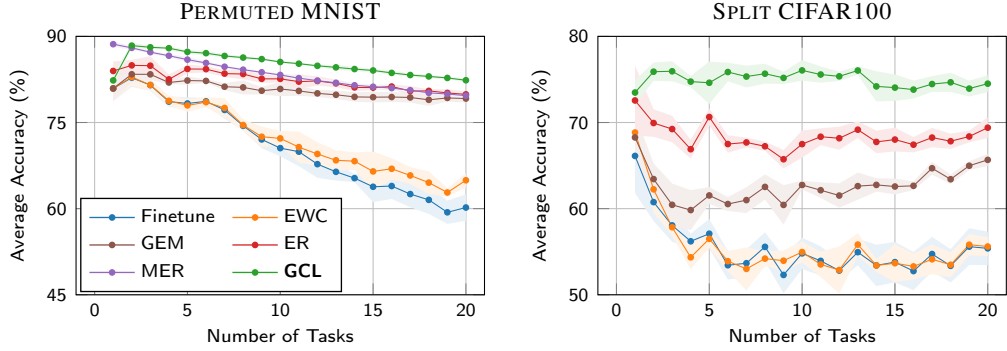

Figure 3: Average accuracy as a function of the number of tasks trained.

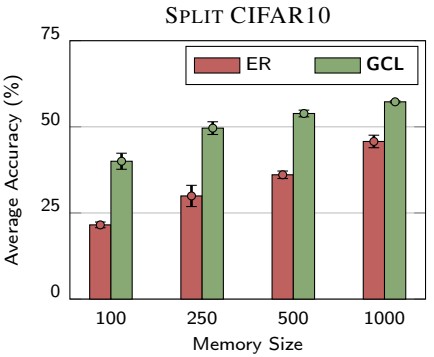

Figure 4: Effects of episodic memory sizes.

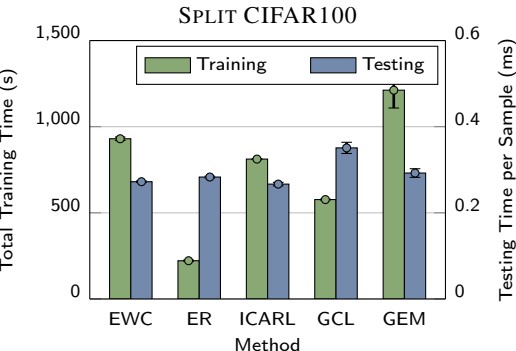

Figure 5: Training and testing time.

**Classification Performance.** Table 1 and 2 show the overall experimental results, and the evolution of performance as a function of the number of tasks are detailed in Figure 3. In every setting, our model (GCL) outperforms the baselines by significant margins, and the gains in performance are especially substantial on complex datasets such as SPLIT CIFAR10 or SPLIT CIFAR100. As noted by Chaudhry et al. (2018b), EWC (Kirkpatrick et al., 2017) performs poorly without multiple passes over the datasets, and we additional find that GEM (Lopez-Paz & Ranzato, 2017) is not very effective under the single-head variants (e.g. on SPLIT SVHN or SPLIT CIFAR10). Task-free approaches such as ER and MER perform more favorably, and such findings are consistent with recent studies (Chaudhry et al., 2019; Riemer et al., 2018).

The advantageous performance of GCL over the other rehearsal strategies can be attributed to its efficient use of the episodic memory. Figure 4 shows that both ER (Chaudhry et al., 2019) and GCL benefit from increases in memory size, but the outperformance of GCL is more visible under the low-resource regime. Sample efficiency, as demonstrated, is especially important since the memory constraints are not relaxable despite the growing complexity of the data distribution during training. It is also worth emphasizing that although our model takes more time to train and evaluate at test time than ER, its training time and testing time are comparable to other approaches (see Figure 5).

**Learned Graphs.** Central to our approach are the pairwise similarities between context images captured by the context graph $\mathbf{G}$. Figure 6 shows a continuous realization of the context graph at the end of continual learning on SPLIT CIFAR10, which has been sorted according to context labels placed underneath the adjacency matrix. Despite being trained exclusively on two classes of target images at a time (e.g., plane & car or bird & cat), the model appears to learn the clustering structure of images relatively well with more pronounced edges formed within classes than across them. The edges across tasks are noisier, but some edges indicate intuitive visual similarities such as those between images of car and truck. We note that the 10-way classification setup in each task encourages the model to clear inter-class edges, especially those within each binary task, so the degree of knowledge transfer across tasks is understandably more subtle.

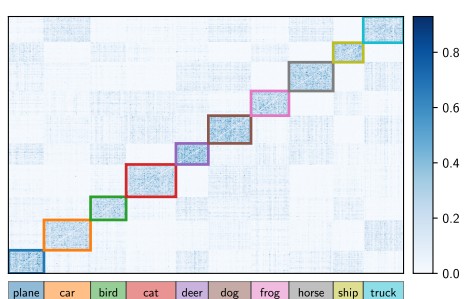

Figure 6: Context graph $\mathbf{G}$ on SPLIT CIFAR10.

Figure 7: Effects of graph regularization ($\lambda_{\mathbf{G}}$).

**Ablation Study.** We further investigate our model performance with an ablation study and summarize it in Table 3. Without the graph regularization term in Equation 7, the model significantly performs worse, indicating that past connections between context samples can help alleviate catastrophic forgetting. By varying the hyper-parameter $\lambda_{\mathbf{G}}$, we also see from Figure 7 that an extreme amount of graph regularization (e.g. $\lambda_{\mathbf{G}} = 1000$) can have detrimental effects on the model performance as well. As alluded earlier, the ability to draw multiple graph samples and average their predictions at test time brings out some gains, as often the case with ensemble methods. Perhaps more importantly, we find that making the context graph $\mathbf{G}$ and the context-target graph $\mathbf{A}$ deterministic results in a dramatic drop in accuracy. The resulting model is a variant of attention mechanism, most similar to attentive neural process (Kim et al., 2019), and as discussed in Section 4, such a deterministic model often relies on a handful of edges, all of which are also prone to distributional shifts and thus catastrophic forgetting as well.

Table 3: Ablation study on SPLIT CIFAR10.

| | | | | |
|---|---|---|---|---|
| Graph regularization | ✓ | ✗ | ✗ | ✗ |
| Multiple graph samples | ✓ | ✓ | ✗ | ✗ |
| Random $\mathbf{G}$ & $\mathbf{A}$ | ✓ | ✓ | ✓ | ✗ |
| Deterministic $\mathbf{G}$ & $\mathbf{A}$ | ✗ | ✗ | ✗ | ✓ |
| Average accuracy | **49.62** | 44.04 | 42.08 | 30.50 |

## 6 CONCLUSION AND DISCUSSION

In this work, we have introduced a graph-based approach to continual learning that exploits pairwise similarities between samples to support knowledge transfer. Based on the learned graphs, we derive a regularization term to guide the training of new tasks against catastrophic forgetting. Our model demonstrates an efficient use of the episodic memory, and as a result, performs competitively under various settings, without requiring access to task definition both during training and at test time in some cases.

As graph-based approaches, including ours, offer a natural way to describe relational inductive biases (Battaglia et al., 2018), we hope that future works further examine the applications of graphs under continual learning settings. If trained well, these graphs can be used not only to share knowledge but also to minimize inference between samples and tasks. A promising direction, for example, is to pose the problem of updating the episodic memory as a graph search and leverage the rich literature on graph theory to devise better strategies for sample selection. As demonstrated by previous works (Aljundi et al., 2019c; Isele & Cosgun, 2018), such selection mechanisms can be effective against catastrophic forgetting, especially when the data distribution is not balanced across tasks.

## 7 ACKNOWLEDGMENT

Financial support is gratefully acknowledged from a Xerox PARC Faculty Research Award, National Science Foundation Awards 1455172, 1934985, 1940124, and 1940276, USAID, and Cornell University Atkinson Center for a Sustainable Future.

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

# A  EXPERIMENT SETUP

We perform experiments on six commonly used classification datasets: PERMUTED MNIST, ROTATED MNIST (LeCun et al., 1998), SPLIT SVHN (Netzer et al., 2011), SPLIT CIFAR10 (Krizhevsky et al., 2009), SPLIT CIFAR100 (Krizhevsky et al., 2009), and SPLIT MINIIMAGENET (Vinyals et al., 2016).

- PERMUTED MNIST (Goodfellow et al., 2013) is a variant of the MNIST dataset of handwritten digits (LeCun et al., 1998), where each task applies a fixed random pixel permutation to the original dataset. The benchmark dataset consists of 20 tasks, each with 1000 samples from 10 different classes.

- ROTATED MNIST (Lopez-Paz & Ranzato, 2017) is another variant of the MNIST dataset of handwritten digits (LeCun et al., 1998), where each task applies a fixed random image rotation to the original dataset. The benchmark dataset consists of 20 tasks, each with 1000 samples from 10 different classes.

- SPLIT SVHN is a variant of the SVHN dataset (Netzer et al., 2011) that consists of 5 tasks, each with two consecutive classes. Since the benchmark dataset is much more challenging than the MNIST variants, we use all of its 73,257 training samples (i.e. 14,650 samples per task) to train our model and the baselines.

- SPLIT CIFAR10 is a variant of the CIFAR-10 dataset (Krizhevsky et al., 2009). Similar SPLIT SVHN, the benchmark dataset consists of 5 tasks, each with two consecutive classes. We use all of its 50,000 training samples (i.e. 10,000 samples per task) to train our model and the baselines.

- SPLIT CIFAR100 is a variant of the CIFAR-100 dataset (Krizhevsky et al., 2009). The benchmark dataset consists of 20 tasks, each with 5 consecutive classes. We use all of its 50,000 training samples (i.e. 2,500 samples per task) to train our model and the baselines.

- SPLIT MINIIMAGENET is a variant of the MINIIMAGENET dataset (Krizhevsky et al., 2009). The benchmark dataset consists of 20 tasks, each with 5 consecutive classes. We use all of its 50,000 training samples (i.e. 2,500 samples per task) to train our model and the baselines. Each image is resized to $84 \times 84$ pixels.

# B  MODEL ARCHITECTURES

As mentioned, while most of previous work uses multi-head architectures and assumes knowledge of task boundaries at test time, we employ a shared classifier head for all tasks. For the MNIST datasets, the image encoders $f_{\theta_1}$ (for graph construction) and $f_{\theta_2}$ (for latent computation) share a multi-layered perceptron with two hidden layers of 256 ReLU neurons, followed by two separate linear mappings, one for each of the encoders. For SPLIT SVHN, SPLIT CIFAR10, SPLIT CIFAR100, and SPLIT MINIIMAGENET, the image encoders share a simple convolutional network with the following structure: `conv 64` $\to$ `conv 64` $\to$ `maxpool` $\to$ `conv 64` $\to$ `conv 64` $\to$ `maxpool` $\to$ `conv 64` $\to$ `conv 64` $\to$ `maxpool`, where `conv NF` is a $3 \times 3$ convolution with `NF` output filters, BatchNorm, and ReLU activations. For all datasets, another linear mapping follows the image encoder $f_{\theta_1}$ before a Gaussian kernel computes the similarities between image embeddings. Finally, the classifier head consists of a RELU activation and a single linear mapping.

# C  BASELINE ARCHITECTURES

We use the same neural network architectures for all the baselines described in the paper: a multi-layered perceptron with two hidden layers of 400 ReLU neurons on PERMUTED MNIST and ROTATED MNIST, following (Hsu et al., 2018), and a ResNet-18 (He et al., 2016) with 20 filters across all layers on other datasets, following (Lopez-Paz & Ranzato, 2017). For all datasets, the baselines consist of more parameters than our corresponding models (see Table 4 for more details).

Table 4: Number of trainable parameters in continual learning models.

| Method | Finetune | EWC | GEM | ER | MER | **GCL** |
|---|---|---|---|---|---|---|
| SPLIT MNIST | 478K | 478K | 478K | 478K | 478K | 406K |
| PERMUTED MNIST | 478K | 478K | 478K | 478K | 478K | 406K |
| ROTATED MNIST | 478K | 478K | 478K | 478K | 478K | 406K |
| SPLIT SVHN | 1.09M | 1.09M | 1.09M | 1.09M | - | 326K |
| SPLIT CIFAR10 | 1.09M | 1.09M | 1.09M | 1.09M | - | 326K |
| SPLIT CIFAR100 | 1.09M | 1.09M | 1.09M | 1.09M | - | 326K |
| SPLIT MINIIMAGENET | 1.09M | 1.09M | 1.09M | 1.09M | - | 343K |

We adopt the implementations of EWC (Kirkpatrick et al., 2017), GEM (Lopez-Paz & Ranzato, 2017), and MER (Riemer et al., 2018) from the authors' repositories [1] [2].

## D   ADDITIONAL TASK-FREE BASELINES

We also note that despite our attempts to tune parameters for MER (Riemer et al., 2018) on SPLIT SVHN and SPLIT CIFAR10, the baseline does not perform reasonably well. The model uses a batch size of 1 and requires multiple passes through the episodic memory per batch, so it is much slower than our model and all other baselines. Due to limited time and computational resources, we do not further investigate the baseline and therefore avoid reporting immature results for fairness.

However, we include results of CN-DPM (Lee et al., 2020), a competitive task-free model based on Dirichlet process mixture models in Table 5. Our setup for SPLIT CIFAR10 is analogous to that of Lee et al. (2020), so we directly quote the numbers for CN-DPM from the paper. Although CN-DPM performs favorably among task-free approaches to continually learning, including GSS (Aljundi et al., 2019c), our model outperforms CN-DPM by a significant margin, even when using a smaller memory size.

Table 5: GCL results and CN-DPM results with different memory sizes.

| Method | SPLIT SVHN | | SPLIT CIFAR10 | |
|---|---|---|---|---|
| | 250 | 500 | 500 | 1000 |
| ER (Chaudhry et al., 2019) | $45.51 \pm 3.03$ | $57.51 \pm 2.77$ | $36.08 \pm 1.09$ | $45.75 \pm 1.82$ |
| CN-DPM (Lee et al., 2020) | – | – | $43.07 \pm 0.16$ | $45.21 \pm 0.18$ |
| **GCL** (Ours) | $\mathbf{60.68} \pm 1.67$ | $\mathbf{65.79} \pm 1.54$ | $\mathbf{53.87} \pm 0.97$ | $\mathbf{57.26} \pm 0.28$ |

## E   MEMORY USAGE

Both GCL and ER (Chaudhry et al., 2019) uses an episodic memory to store images and labels from past tasks. The only additional memory usage of GCL comes from the context graph $\mathbf{G}$, which is represented by a square matrix whose entries intuitively describe pairwise similarities between such images. Given a memory consisting of $|\mathcal{M}|$ images of size $C \times H \times W$, it only requires $|\mathcal{M}|^2$ floating points to store the matrix.

Table 6: Memory usage of ER and GCL for various datasets.

| DATASET | $|\mathcal{M}|$ | Image Size | ER | GCL |
|---|---|---|---|---|
| PERMUTED MNIST | 1000 | $1 \times 28 \times 28$ | 3.284 MB | 7.284 MB |
| ROTATED MNIST | 1000 | $1 \times 28 \times 28$ | 3.284 MB | 7.284 MB |
| SPLIT CIFAR10 | 250 | $3 \times 32 \times 32$ | 3.109 MB | 3.359 MB |
| SPLIT SVHN | 250 | $3 \times 32 \times 32$ | 3.109 MB | 3.359 MB |
| SPLIT CIFAR100 | 500 | $3 \times 32 \times 32$ | 6.219 MB | 7.199 MB |
| SPLIT MINIIMAGENET | 500 | $3 \times 84 \times 84$ | 42.408 MB | 43.389 MB |

---

[1] https://github.com/facebookresearch/GradientEpisodicMemory
[2] https://github.com/mattriemer/mer

As seen from Table 6, the memory usage of GCL are very similar the same as that of ER, except when both are very small as in the case of PERMUTED MNIST and ROTATED MNIST, because (1) continual learning algorithms are often required to use a very small $|\mathcal{M}|$ and (2) the cost for storing natural images are often much higher than that of the context graph.

As the number of tasks increases, it is perhaps essential to expand the episodic memory, in which case the quadratic growth of the latter might dominate the linear increase of the former (e.g. $|\mathcal{M}| = 5000$ and images are of size $3 \times 32 \times 32$). Although we have not practically encountered such a problem with GCL, we note that the quadratic growth of the number of entries in the context graph can be reduced to a linear growth in memory requirements. More specifically, each entry is the output of the kernel function $\kappa_\tau$ (see Section 3, e.g. $\kappa_\tau(\mathbf{u}_i, \mathbf{u}_j) = \exp\left(-\frac{\tau}{2}\|\mathbf{u}_i - \mathbf{u}_j\|_2^2\right)$), so we could easily store $|\mathcal{M}|$ intermediate embeddings $\{\mathbf{u}_i\}$ at each step and apply the kernel function on the fly, which is especially beneficial when $\mathbf{u}_i$ are much lower dimensional than the original images.

## F  ADDITIONAL EXPERIMENT RESULTS

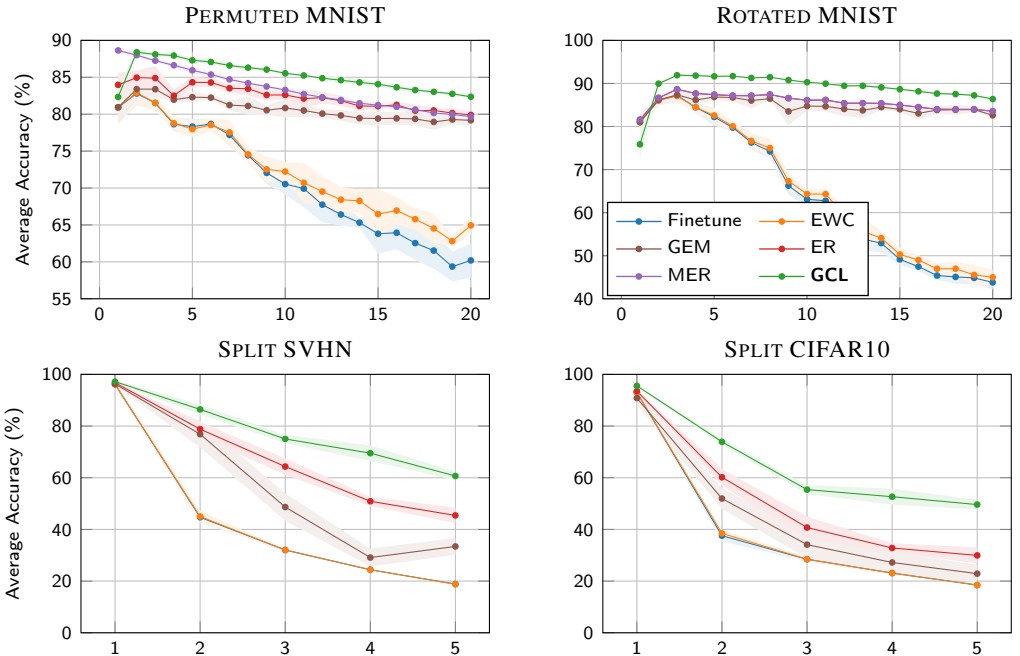

Figure 8: Average accuracy as a function of the number of tasks trained on PERMUTED MNIST, ROTATED MNIST, SPLIT SVHN, and SPLIT CIFAR10.

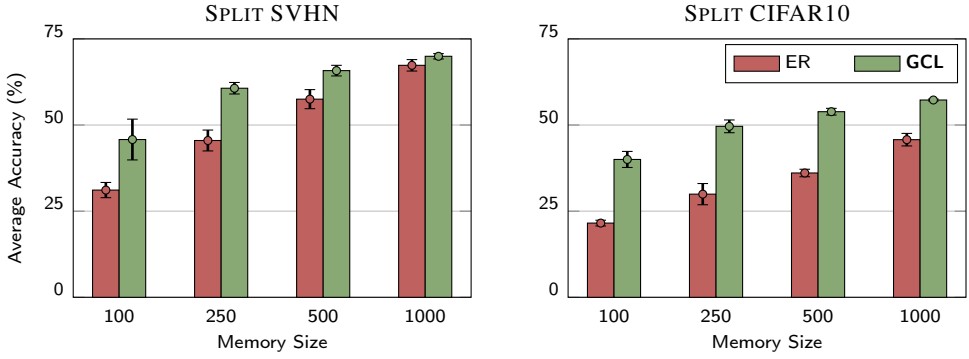

Figure 9: Average accuracy as a function of memory size on SPLIT SVHN and SPLIT CIFAR10.

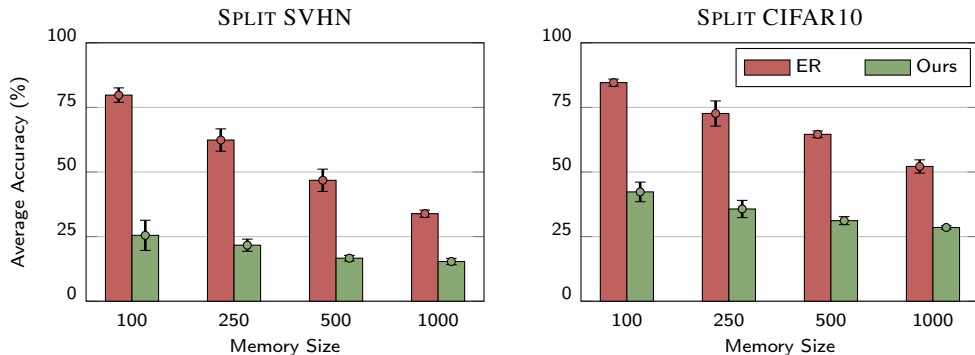

Figure 10: Average forgetting as a function of memory size on SPLIT SVHN and SPLIT CIFAR10.

## G  HYPER-PARAMETERS

Following Lopez-Paz & Ranzato (2017), we report the hyper-parameter grids considered in our experiments. These hyper-parameters are selected independently for each model, and the best values are given in parenthesis.

- Finetune
  - optimizer: [Adam (Split SVHN, Split CIFAR10), SGD (Permuted MNIST, Rotated MNIST)]
  - learning rate: [0.0002, 0.001 (Split SVHN, Split CIFAR10, Split CIFAR100, Split MiniImagenet), 0.01, 0.1 (Permuted MNIST, Rotated MNIST), 0.3, 1.0]
- EWC (Kirkpatrick et al., 2017)
  - optimizer: [Adam, SGD (Permuted MNIST, Rotated MNIST, Split SVHN, Split CIFAR10)]
  - learning rate: [0.0002 (Split SVHN, Split CIFAR10), 0.001, 0.01, 0.1 (Permuted MNIST, Rotated MNIST), 0.3, 1.0]
  - regularization: [0.1, 1, 10 (Permuted MNIST, Rotated MNIST), 100 (Split SVHN, Split CIFAR10, Split CIFAR100, Split MiniImagenet), 1000]
- GEM (Lopez-Paz & Ranzato, 2017)
  - optimizer: [Adam (Split SVHN, Split CIFAR10), SGD (Permuted MNIST, Rotated MNIST)]
  - learning rate: [0.0002, 0.001 (Split CIFAR10, Split CIFAR100, Split MiniImagenet), 0.01, 0.1 (Permuted MNIST, Rotated MNIST), 0.3, 1.0]
  - margin: [0.0, 0.1, 0.5 (Permuted MNIST, Rotated MNIST), 1.0 (Split SVHN, Split CIFAR10, Split CIFAR100, Split MiniImagenet)]
- MER (Riemer et al., 2018)
  - optimizer: [SGD (Permuted MNIST, Rotated MNIST)]
  - learning rate: [0.0002, 0.001, 0.01, 0.1 (Permuted MNIST, Rotated MNIST), 0.3, 1.0]
  - within batch meta-learning rate (beta): [0.01 (Permuted MNIST, Rotated MNIST, Split CIFAR10, Split CIFAR100, Split MiniImagenet), 0.03, 0.1, 0.3, 1]
- ER (Chaudhry et al., 2019)
  - optimizer: [Adam (Split SVHN, Split CIFAR10, Split CIFAR100, Split MiniImagenet), SGD (Permuted MNIST, Rotated MNIST)]

- learning rate: `[0.0002, 0.001 (Split SVHN, Split CIFAR10, Split CIFAR100, Split MiniImagenet), 0.01, 0.1 (Permuted MNIST, Rotated MNIST), 0.3, 1.0]`

- GCL
  - optimizer: `[Adam (Permuted MNIST, Rotated MNIST, Split SVHN, Split CIFAR10, Split CIFAR100, Split MiniImagenet), SGD]`
  - learning rate: `[0.0002, 0.001 (Permuted MNIST, Rotated MNIST, Split SVHN, Split CIFAR10, Split CIFAR100, Split MiniImagenet), 0.01, 0.1, 0.3, 1.0]`
  - graph regularization: `[0, 10, 50 (Split SVHN, Split CIFAR10, Split CIFAR100, Split MiniImagenet), 100, 1000, 5000 (Rotated MNIST)]`
  - context temperature: `[0.1 (Permuted MNIST, Rotated MNIST), 0.3, 1 (Split SVHN, Split CIFAR10, Split CIFAR100, Split MiniImagenet), 5, 10]`
  - target temperature: `[0.1, 0.3, 1, 5 (Permuted MNIST, Rotated MNIST, Split SVHN, Split CIFAR10, Split CIFAR100, Split MiniImagenet), 10]`

