# OpenReview forum: "Graph-Based Continual Learning"
_ICLR.cc/2021/Conference — ICLR 2021 Spotlight_

### Official Review · AnonReviewer1 · 2020-10-28
**Novel idea of using random graphs for continual learning**

**Rating:** 7
**Confidence:** 4

**Review:**

Summary:

The paper proposes a novel way of using random graphs to improve task-free continual learning method. It builds to random graphs, G and A, based on the similarity of images stored in the memory and those of the current tasks, and utilize the relative information to build representation of the images and predict. The idea is well-formulated, and carried out in a sound way. The graph regularization term resembles the knowledge-distillation, as the authors also mentioned, but it serves different purpose of preserving the covariance structure of the outputs of the image encoders.

The experimental results look quite strong and the ablation study also looks sound. Overall, I find the paper quite strong.

Con & Questions:

- It seems like there are 4 hyperparameters: "temperature" for the Gumbel-Softmax relaxation and 3 lambdas. Fig 6 shows the effects of $\lambda_G$, but what about others? How sensitive is the performance with respect to other hyperparamters? Since the problem setting is a single-pass setting, how are the hyperparameters selected?

- The inference time is a bit longer (not too much than others) since the algorithm has to sample graphs 30 times. What happens if the number of samples is less than 30 so that the inference time becomes similar to ER? Would the performance degrade significantly? How critical is the sample size 30?

---

> ### Author Response · Authors · 2020-11-19
> **Hyper-pameter Selection and Number of Graph Samples**
>
> We thank the reviewer for the detailed review and encouraging feedback. Our responses to the questions are below.
>
> * **Hyper-parameter Selection**. We don’t do exhaustive search for the hyper-parameters and opt for common-sense values in all of our experiments. For $\lambda_{\text{C}}$ and $\lambda_{\text{T}}$ (see Equation 8 for their definitions), we simply set both of them to $1.0$, effectively weighting old samples and new samples equally. As mentioned in the paper, we use a small temperature for $\mathbf{G}$ (e.g. $\tau_{\mathbf{G}} = 1.0$)  and a larger temperature for $\mathbf{A}$ (e.g. $\tau_{\mathbf{A}} = 5.0$) to avoid spurious edges within the context graph and encourage more sample-to-sample association at test time. We show the effects of $\lambda_{\mathbf{G}}$ in Figure 6 to demonstrate that including graph regularization is particularly helpful to our model performance. Please see Appendix G for more details about the hyper-parameters.
>
> * **Number of Graph Samples**. The following table shows the effects of the number of graph samples $\mathbf{A}$ on the accuracy of our model on Split CIFAR10. Even with a single graph sample, our model performs relatively well compared to ER ($44.38 \%$ vs $29.94\%$). As the number of graph sample increases, the classification performance gets better at a diminishing rate. We find that 30 samples work quite well without incurring too much overhead at test time.
>
>     $$\\begin{array} {|l|c|c|c|c|}\\hline  \\text{Graph Samples} & 1 & 10 & \\textbf{30} & 50 \\\\ \\hline \\text{Accuracy (ACC)} & 44.38 \\pm 1.25 & 46.69 \\pm 1.78 & \\textbf{49.62} \\pm 1.85 & 49.67 \\pm 1.44\\\\ \\hline  \\end{array}$$

---

### Official Review · AnonReviewer2 · 2020-10-28
**Interesting paper**

**Rating:** 8
**Confidence:** 4

**Review:**

This paper presents an interesting idea of using random graphs to represent relational structures amongst contextual samples and between contextual samples and target samples. Besides, the authors propose a regularization objective to alleviate catastrophic forgetting.

The novelty mainly comes from the use of graph structure to preserve the memory of previous tasks. The results on public datasets are encouraging. Also, the authors show some correspondence between the learned graphs and underlying (clustering) structure of data. It would be great if the authors can also extend this algorithms to other real-world datasets, e.g., capturing the climate or environmental changes over time, and provide more interpretation of learned graph structrure.

---

> ### Author Response · Authors · 2020-11-19
> **Extension to Environmental Datasets**
>
> We would like to thank the reviewer for the positive assessment of the paper. We agree that it would be interesting to test our model on other real-world datasets as suggested by the reviewer, but due to limited time and scope, we will probably leave it for future work.

---

### Official Review · AnonReviewer4 · 2020-10-28
**an interesting integration of continual learning, random graph, and attention network**

**Rating:** 7
**Confidence:** 3

**Review:**

This paper presented a memory-based continual learning model where relationships between training samples are represented with a random graph that is defined from the non-linear embedding of the input data. Catastrophic forgetting between tasks is partially (1) alleviated with a graph regularization that penalizes changes of random graph statistics, and (1) memory replay and reservoir sampling to update memory. The performance of this model is evaluated against several state-of-the-art models to handle the catastrophic loss.

I tend to rate this paper as a good paper for acceptance because the presentation is clear, the idea is novel and the experiment is convincing.

Pros:

+ clear presentation in Section 3, Graph-Based Continual Learning, plus Figure 1.
+ Good literature survey in Section 4, Related Work.
+ A good idea to related graph network and attention mechanism.

Cons:

- Are all edges equally important in the regularisation term in Eq. 7? I feel more theory could be dug out by using, for example, tangent distance.
- What does a graph representation really give us, since as authors mentioned, this is in a sense an attention mechanism? Can higher-order graph relationships be utilized if we replace a random graph with a ergm containing more structures?

---

> ### Author Response · Authors · 2020-11-19
> **Edge Importance in Graph Regularization and Higher-Order Graph Structures**
>
> We thank the reviewer for the detailed review and suggestions. Our responses to the questions are below.
>
> * **Edge Importance in Graph Regularization**. For simplicity, we treat the edges in Equation 7 the same way. However, the cross-entropy loss that we use to penalize deviations from learned edges does account for edge importance to some extent. Given a previously learned edge $\text{Ber}(p)$ and its current representation $\text{Ber}(q)$, the cross-entropy loss is $H(p, q) = -p \log(q) - (1-p) \log(1 - q)$, which is an increasing function of $p$ when $q \leq 0.5$. As a result, if the model returns the same probability $q \leq 0.5$ for some edges, the edge with the smallest learned probability contributes the least to the regularization term (the reverse is true when $q \geq 0.5$). Conceivably, the model tends to learn sharp edges (those with $p$ far away from $0.5$), which agrees with our empirical observation that a sparse context graph $\mathbf{G}$ results in better accuracy.
>
> * **Higher-Order Graph Structures**. In this paper, we only consider pairwise connections between samples in the form of adjacency matrices. It would be a really interesting direction to model and learn higher-order relationships (e.g. triples, quadruples, etc.) as suggested by the reviewer. We think that such learned structures would provide further signals for sample selection strategies, which can be effective against catastrophic forgetting as demonstrated by previous work [1, 2].
>
> 1. Aljundi, R., Lin, M., Goujaud, B. and Bengio, Y., 2019. Gradient based sample selection for online continual learning. In Advances in Neural Information Processing Systems (pp. 11816-11825).
> 2. Isele, D. and Cosgun, A., 2018. Selective experience replay for lifelong learning. arXiv preprint arXiv:1802.10269.

---

### Official Review · AnonReviewer3 · 2020-10-29
**The paper provides a new perspective for replay-based continual learning. Unlike the traditional replay method, this paper uses a learnable random graph to expand this memory. Use graphs to capture pairwise similarities between samples. The model can not only prevent forgetting, but also help the learning of new tasks. Good results have been achieved through the given experimental results.**

**Rating:** 6
**Confidence:** 4

**Review:**

Strengths:
 - the combination of components is novel, Especially, this paper consider the relevance of instance in memory.
- The model can not only prevent forgetting, but also help the learning of new tasks. The promotion of current tasks is also crucial in continual learning.
Weaknesses:
- the empirical validation is weak. Therefore, more new models need to be compared. For more details, please refer to “Reasons for reject”

Reasons for accept:
1.	The structure of this paper is clear and easy to read. Specifically, the motivation of this paper is clear and the structure is well organized; the related work is elaborated in detail; the experimental setup is complete.
2.	Based on the use of replay to solve catastrophic forgetting, the current popular graph structure is introduced to capture the similarities between samples. Combined with the proposed Graph Regularization, this paper provides a new perspective for solving catastrophic forgetting.
3.	The experimental results given in the paper can basically show that the proposed method is effective. The ablation study also verified the effectiveness of each component.

Reasons for reject:
1.	The lack of comparison of experimental effects after replacing Graph Regularization with other regularization methods mentioned in this paper, or other distance measurement methods, eg., L2.

2.	This paper compares relatively few baselines, especially recent studies. I hope to see the comparison results of some papers in the list below. The latest papers on the three types of methods (regularization, expansion, and rehearsal) for solving catastrophic forgetting are included. Therefore, if it can be compared with some of these models, it will be beneficial to the evaluation of GCL.

[1] Ostapenko O , Puscas M , Klein T , et al. Learning to Remember: A Synaptic Plasticity Driven Framework for Continual Learning. ICML 2019
[2] Y Wu, Y Chen, et al. Large Scale Incremental Learning. CVPR 2019
[3] Liu Y , Liu A A , Su Y , et al. Mnemonics training: Multi-class incremental learning without forgetting. CVPR 2020
[4] Zhang J , Zhang J , Ghosh S , et al. Class-incremental learning via deep model consolidation. 2020 IEEE Winter Conference on Applications of Computer Vision (WACV)
[5] Guanxiong Zeng, Yang Chen, Bo Cui, and Shan Yu. Continuous learning of context-dependent processing in neural networks. Nature Machine Intelligence, 2019.
[6] Wenpeng Hu, Zhou Lin, et al. Overcoming catastrophic forgetting for continual learning via model adaptation. ICLR 2019
[7] Rao D , Visin F , Rusu A A , et al. Continual Unsupervised Representation Learning. NeurIPS 2019

---

> ### Author Response · Authors · 2020-11-19
> **Comparison to Regularization Models and Other Baselines**
>
> We thank the reviewer for the constructive comments and especially the recent references. Our responses to the questions are below.
>
> * **Comparison to Regularization Models**. In the paper, we did compare our model with EWC, one of the most popular regularization approaches. Here, we also compare it with VCL[1] based on [its authors’ public repository](https://github.com/nvcuong/variational-continual-learning). As shown in previous work [2], regularization techniques such as EWC or VCL are not very effective in single-pass settings.
>
>     $$\\begin{array} {|l|c|c|}\\hline  \text{Dataset} & \text{Permuted MNIST} & \text{Rotated MNIST} \\\\ \\hline \text{EWC} & 64.94 \pm 1.22 & 44.99 \pm 1.73 \\\\ \\hline  \text{VCL} & 64.50 \pm 1.78 & 49.13 \pm 1.33 \\\\ \\hline   \textbf{GCL} & \textbf{82.36} \pm 0.36 & \textbf{86.37} \pm 0.32 \\\\ \\hline \\end{array} $$
>
> * **Comparison to Other Baselines**.  We appreciate the recent studies pointed out by the reviewer and have updated our related work section to include these ones. While these models are impressive, we emphasize that they are not directly comparable to our approach and often assume unlimited access to datasets [3, 4, 5, 6, 7, 8] or pre-trained feature extractors [6]. In contrast, we focus on single-epoch settings where each task is experienced only once, following [9, 10, 11, 12]. As discussed in the paper and argued in previous work, such settings are much more challenging and are designed to expose catastrophic forgetting. For example, a naive MLP trained continuously on Rotated MNIST (denoted by *Finetune* in the paper) can significantly increase average accuracy from $43.80$ after the first epoch to $62.42$ after 5 epochs to $80.99$ after 20 epochs.
>
>
> 1. Nguyen, C.V., Li, Y., Bui, T.D. and Turner, R.E., 2017. Variational continual learning. arXiv preprint arXiv:1710.10628.
> 2. Chaudhry, A., Ranzato, M.A., Rohrbach, M. and Elhoseiny, M., 2018. Efficient lifelong learning with a-gem. arXiv preprint arXiv:1812.00420.
> 3. Ostapenko, O., Puscas, M., Klein, T., Jahnichen, P. and Nabi, M., 2019. Learning to remember: A synaptic plasticity driven framework for continual learning. In Proceedings of the IEEE Conference on Computer Vision and Pattern Recognition (pp. 11321-11329).
> 4. Wu, Y., Chen, Y., Wang, L., Ye, Y., Liu, Z., Guo, Y. and Fu, Y., 2019. Large scale incremental learning. In Proceedings of the IEEE Conference on Computer Vision and Pattern Recognition (pp. 374-382).
> 5. Zhang, J., Zhang, J., Ghosh, S., Li, D., Tasci, S., Heck, L., Zhang, H. and Kuo, C.C.J., 2020. Class-incremental learning via deep model consolidation. In The IEEE Winter Conference on Applications of Computer Vision (pp. 1131-1140).
> 6. Zeng, G., Chen, Y., Cui, B. and Yu, S., 2019. Continual learning of context-dependent processing in neural networks. Nature Machine Intelligence, 1(8), pp.364-372.
> 7. Hu, W., Lin, Z., Liu, B., Tao, C., Tao, Z., Ma, J., Zhao, D. and Yan, R., 2018, September. Overcoming catastrophic forgetting for continual learning via model adaptation. In International Conference on Learning Representations.
> 8. Rao, D., Visin, F., Rusu, A., Pascanu, R., Teh, Y.W. and Hadsell, R., 2019. Continual unsupervised representation learning. In Advances in Neural Information Processing Systems (pp. 7647-7657).
> 9. Lopez-Paz, D. and Ranzato, M.A., 2017. Gradient episodic memory for continual learning. In Advances in neural information processing systems (pp. 6467-6476).
> 10. Riemer, M., Cases, I., Ajemian, R., Liu, M., Rish, I., Tu, Y. and Tesauro, G., 2018. Learning to learn without forgetting by maximizing transfer and minimizing interference. arXiv preprint arXiv:1810.11910.
> 11. Chaudhry, A., Rohrbach, M., Elhoseiny, M., Ajanthan, T., Dokania, P.K., Torr, P.H. and Ranzato, M., 2019. Continual learning with tiny episodic memories.
> 12. Aljundi, R., Lin, M., Goujaud, B. and Bengio, Y., 2019. Gradient based sample selection for online continual learning. In Advances in Neural Information Processing Systems (pp. 11816-11825).

---

### Decision · Program_Chairs · 2021-01-07
**Final Decision**

**Decision:**

Accept (Spotlight)

**Comment:**

This paper presents an interesting idea for task-free continual learning, which makes use of random graphs to represent relational structures among contextual and target samples. The reviewers agreed that the technical idea is novel, the experiments are extensive and the presentation is good. The authors addressed the reviewers' concerns in the rebuttal. I recommend to accept.